# Fire-Temperature Influence on Portland and Calcium Sulfoaluminate Blend Composites

**DOI:** 10.3390/ma13225230

**Published:** 2020-11-19

**Authors:** Konrad A. Sodol, Łukasz Kaczmarek, Jacek Szer

**Affiliations:** 1Institute of Materials Science and Engineering, Lodz University of Technology, 90-924 Łódź, Poland; Lukasz.Kaczmarek@p.lodz.pl; 2Department of Building Physics and Building Materials, Lodz University of Technology, 90-924 Łódź, Poland; Jacek.Szer@p.lodz.pl

**Keywords:** fire, calcium sulfoaluminate cement, Portland cement

## Abstract

This paper presents the research data of the fire-temperature influence on Portland CEM I (OPC) and calcium sulfoaluminate (CSA) types of cement blend composites as cooling materials dedicated for infill and covers in fire systems. The data present the material responses for four types at high-temperature elevation times (0, 15, 30, 60 min), such as core heat curves, differences in specimens color, flexural and compressive strength parameters. Materials were tested using the DSC method to collect information about enthalpies. The differences between cement blend composites were compared with commonly used cooling materials such as gypsum blends. It is shown that modifications to Portland cement composites by calcium sulfoaluminate cement have a significant influence on the cooling performance during high-temperature, even for 60 min of exposure. The temperature increase rates in the material core were slower in composites with regards to additionally containing calcium sulfoaluminate in 100–150 °C range. After 60 min of high-temperature elevation, the highest flexural and compressive strength was 75% OPC/25% CSA cement composition. The influence on cooling properties was not related to strength properties. The presented solution may have a significant influence as a passive extinguisher solution of future fire resistance systems in civil engineering.

## 1. Introduction

Safety concerns during a fire are one of the highest challenges for Civils Engineers today [1]. Chapter VI of Announcements of the Minister of Infrastructure and Development since 16 July 2015, as amended [2], deals with fire-building protection. By law, every building must be designed and built in a way restrictive fire spreading and giving a possibility to rescue operations. Civil objects are divided into five fire classes depending on the purpose of the object, building rise and a number of stories. Classes are described as fire resistance requirements to be met by internal and external parts of a building. Internal walls such as partition walls can be made from bricks, drywall [3] or glass partition, which is gaining popularity [4]. Fire Resistance requirements for these elements are from EI 15 to EI 60 class. External facades should meet requirements from EI 30 to EI 120 (o↔i). Fire-temperature is particularly unfavorable for building joineries like windows, doors, partition and facades made of materials with low melting point and high thermal conductivity, for instance, aluminum. Temperature increases cause strength properties to decrease [5,6]. To meet the demanding Fire Resistance class requirements, additional infill and/or cover insulation and cooling materials are needed [7,8]. These solutions are presented in the figures below.

As presented in Figure 1, there are few possibilities to protect metal profiles against dangerous fire-temperature. In Facades, doors, windows, partitions, etc., where the esthetical point of view is needed supporting cooling-insulation materials are filling profile (Figure 1a,b). Sometimes these materials are inside another profile (Figure 1a), which is inserted as a reinforcement—cooling profile into the standard façade profile. Typical contour protection is based on board fixed to external profile layers. This is the most common solution in the fire protection of steel structures. Protection against fire is strictly connecting with the thermal conductivity of protected material [9]. Materials with good thermal conductivity lose their load-bearing properties during exposure to high-temperature quickly. That is the reason why additional isolation material from the external side of the profile is needed [8]. If project demands do not allow additional external covers, for instance, in façade profiles, then supporting material must have a cooling function and be installed inside the profile in chambers. If the profile has one chamber, then the insulation function in chamber space is useless. If the profile has more than one (Figure 1b), then a combination of insulation and cooling materials may meet the standard requirement. The increasing temperature during fire-temperature causes decreases in strength properties metal profiles. That is the reason why more useful supporting materials should base on reaction connected with heat energy consumption. There are two main mechanisms of heat energy consumption. The first is based on specific heat capacity, in which the delivered amount of energy, in the form of heat to one unit of the mass of the substance, causes an increase of one unit of temperature. The second is based on an endothermic reaction, in which the delivered temperature will increase the enthalpy of the endothermic reaction and cools down the system [10].

In the paper [11], it is pointed out that the main goal of using heat absorption coating is decreasing heat flux transported to supporting structure. The most popular fire protection infill materials are gypsum board, calcium silicate and silicate–cement materials [12]. Exposition to the high-temperature is leading to physicochemical changes for most of the cement and gypsum-based materials [13,14,15]. Changes cause, for instance, a decrease in strength properties. In the paper [16], thermodynamic changes occur in the matrix and in aggregates [17], for example, a quartz phase transition to cristobalite following the increase of volume [18], even 17% [19] or carbonate decomposition with CO_2_ emission [20] in limestone. Cement materials are described as engineer construction material [14] also used in industry as special thermal resistance material [21,22]. Heat treatment cements material according to the heat-resistance standard, has a different influence on the material than exposure on fire-temperature in according to the standard fire curve [23]. In the hardened Portland matrix, there are more than 30 phases, of which 7th are most common [24]. There is also the possibility to create a new phase as a consequence of an interaction between aggregate, cement phases and water [25]. In the reaction of Portland cement with water, there are hydrate aluminous, calcium–silicate and ferrous phases. If gypsum is added as the bonding agent, then sulfate phases will hydrate. The main phases are hydrated calcium–silicate (CSH) and calcium hydroxide (CH), brownmillerite (C2AF) and ettringite (C4AH13) are smaller share phases [26].

High-temperature exposure causes moisture and capillary capture water retention, dehydration from hydrates [16] and dehydroxylation of hydro oxygens like portlandite. Every phase has a characteristic phase decomposition temperature range, according to Table 1.

Portland cement usage includes high-temperature composites for work in <400 °C [21] after heat treatment. Fast temperature increase can cause a spalling effect [16,28]. Another type of binder is calcium sulfoaluminate cement. This material has a different composition of clinker than Portland cement. Hydration reaction is different than for Portland cement. Consequently, hydration is gaining an expansive hydraulic binder [29], where the main phase is ettringite. Additional phases are monosulfite, stratlingite and alumina trihydrate [30]. Phase decompositions underexposure on the high-temperature are shown in Table 2 [31].

CSA types of cement show a high permeability and nowadays they are dedicated for work at temperatures lower than 150 °C [32]. Materials based on CSA cement are characterizing by higher compressive strength than traditional types of Portland cement (ex. 42,5R/52,5R) [29]. OPC-based material modified by CSA influences hardening time reduction, early strength and standard strength [33]. OPC shrinkage behavior during drying is one of the biggest issues conducting to cracking and curling. This negative behavior can be compensated by using expansive or shrinkage-compensating concrete bases, for instance, by CSA cement addition, where properties are strongly connected with types of curing [34]. On the other hand, reverse modification has a positive influence on steel passivation [35] by setting phase able to carbonation.

The main ingredient of gypsum is calcium sulfonate dehydrate—gypsum stone. This compound is classified according to water content into three types: dihydrate, hemihydrate (α i β) and anhydrate [36]. Gypsum meets fire resistance standard requirements because it is classified as non-flammable [37]. The use of gypsum in fire protection systems complies with the standard PN-EN 13501-1. It is A2 class material classified as noncombustible and no flashover in RCT referential test [38]. In fire zones, it is used as byproducts of a matrix in fire resistance gypsum board reinforced by cut glass fiber. High-temperature exposure dehydrates gypsum, and next heat energy is consumed on water evaporation in the following reaction [39]:$${\mathrm{CaSO}}_{4}\cdot 2H_{2}O{-}^{100{}^{\circ}C}\to {\mathrm{CaSO}}_{4}\cdot \frac{1}{2}H_{2}O{-}^{200℃}\to {\mathrm{CaSO}}_{4}$$

Water released from crystal lattice has an extinguishing function, which can decrease the external field of temperature [40]. Based on thermodynamic calculations, it is possible to get further decomposition stage of gypsum in temperature greater than 800 °C and appropriate pressure in accordance with the following reaction [41]:$${\mathrm{CaSO}}_{4}\to {\;}^{800-1375\;{}^{\circ}C}\to \mathrm{CaO}+{\mathrm{SO}}_{2}+\frac{1}{2}O_{2}$$

Data concerning the decomposition of phases in ceramic material in cement bases are presented in Collier N. C. paper [42]. These were selected for the purposes of this study according to Portland cement, gypsum and CSA cement and shown in the Table 3.

The main solution of cooling properties is to use material composite and internal endothermic reaction to increase the enthalpy and for creating products able to periodically decrease the temperature or keep them on the same level following La Chatelier and Braun principle from 1887. Insulation-cooling properties of ceramic materials on cement-based was not investigated as thoroughly as insulation-construction properties [43] or fire resistance insulate covers [44,45,46]. Critical condition influence like fire on the behavior of composite with CSA contents has not found a place in literature. There is no information about this cement-based material behavior above 150 °C and behavior depending on fire exposure time. There is also a lack of information about OPC-CSA blends exposure to elevated temperature. Available research describes the usage of CSA cement in standard environment conditions on special projects, for example, airport runaway, bridge decks, prestressed tanks, etc. [47]. Furthermore, CSA cement has a binder with a lower carbon footprint than OPC cement [48]. However, in [49] authors pointed important conclusion connected with the general safety of construction material, that life–safety will always take precedence over sustainability issues.

## 2. Materials and Methods

To understand phenomena occurring within the composite material containing CSA cements under elevated temperature, specimens with the following ingredients were created with a weight according to Table 4:OPC—Portland cement OPC CEM I 42.5 R Górażdże PremiumCSA—AliCem ItalicementiBuilding gypsum Dolina NidyWater from waterworksCut glass fiber 4.5 mm

A water/gypsum ratio = 0.6 was forced by manufacturer specification for cement blends water/cement ratio = 0.5 was used. A glass fiber/total binder ratio = 0.02. 12 pcs. 40 × 40 × 160 mm according to standard PN-85/B-04500 was molded from every composition. Each ingredient specimen from the cement group was marked Z(1–5). Each specimen from the chosen group was marked additionally 1–12 according to mold position. Specimens were treated according to PN 85/B-04500 standard. After 28 days, the samples were subjected to subsequent tests. Gypsum specimens were marked G1. Each specimen from this group was marked additionally 1–12 according to mold position as shown in Figure 2. Gypsum specimens were cured in the air for 14 days. After 14 days, the samples were tested.

### 2.1. Samples Behavior during Elevation on High-Temperature

A temperature test was conducted in an atmosphere electrical furnace (Zakład Elektromechaniczny NEOTERM, Wrocław, Poland). The heat conditioning time was equal to the appropriate EI fire resistance class, meaning: 0 min, 15 min, 30 min, 60 min, in accordance with PN-EN 13501. Three samples were placed in the oven (Zakład Eletromechaniczny NEOTERM, Wrocław, Poland). Thermocouples (TYPE K 1200 °C, ∅2.5) were inserted in ∅3x80 holes, as presented in Figure 1. Temperatures were measured during all tests. After the 15/30/60 min tests, specimens were brought out and cooled at laboratory temperature equal to 23 ± 2 °C.

### 2.2. Differential Scanning Calorimetry

To study thermal events and relations in cement pastes during high temperature, we used the DSC method. Analyses were carried out using DSC 204 Netzsch station. Aluminum crucibles (Netzsch, Selb, Deutschland) were used in investigations. All experiments were conducted at a temperature from 25 to 600 °C with a heating rate of 15.0 K/min in a nitrogen atmosphere with a 20.0 mL/min gas flow.

### 2.3. Flexural and Compressive Strength

Strength tests were performing following PN-EN 1015-11 “Methods of test for mortar for masonry. Determination of flexural and compressive strength of hardened mortar” to determine elevated temperature influence on flexural and compressive strength [50]. Samples were testing which meets the standard requirements after heat-treating.

## 3. Results and Discussion

### 3.1. Exposure to High-Temperature

Next, the weight of each specimen was measured, and the average density of specimens was calculated. The results are presented in Table 5.

Density decrease after high-temperature is an expected phenomenon. It is explained in literature as the following phase changes with new products released as a result of heat energy delivery [51]. All samples after furnace test are shown in Figure 3.

After 15 min in the furnace, the gypsum specimens showed cracking of the visible surfaces. Longer time under high-temperature conditioning made the cracking more visible and deep. After 60 min, the gypsum specimens were fully cracked. After 15 min, smaller cracking was observed in the cement specimens than in the gypsum specimens. At 30 min at high-temperatures, cracking was on the same level as at the previous time. The most visible cracking lines were observed at Z1 and G1 samples. Volume expansion of cementous samples was a process observed for a time longer than 30 min. Visible crushing of the top corners and partial edges was visible in samples containing OPC cement under loads greater than 30 min. The corners and edge erosion may have been forced by dehydration and dehydroxylation of the original structure following moisture absorption by free lime during cooling in the air after heat treatment [16]. In parallel, the cracking effect after heating may have happened as an effect of thermal shock as a result of environmental changes from the hot furnace to a much colder laboratory [52]. The conducted tests gave data about the samples’ thermal response to elevated temperatures. The average results are presented in Figure 4.

The graph shows the relationship between each type of cement sample related to gypsum. Z1 composition temperature increased faster in 34 min than gypsum. The gypsum thermal decomposition at 100–150 °C was the longest process. The temperature increased after crossing 150 °C and finally reached 729 °C in 60 min. Between 150 °C and 729 °C, no significant cooling effects were observed. This characteristic temperature interval is also characteristic for ettringite and C-S-H decomposition, as shown in Table 3. CSA increased in the OPC matrix gave an elongated time of temperature increase in the 100–150 °C temperature interval. Between 3:51 min and 11:31 min, the Z1 composite temperature was around 100 °C, which means that ettringite decomposition had stopped the temperature increase. After 11:31 min, the temperature in the Z1 core climbed to 110 °C, and then slowly rose to 150 °C over 26 min. This shows the next phase of decomposition. CSA addition in the OPC matrix lengthened the time of temperature rise in the core samples. An improvement in properties followed the increase in CSA cement. The positive time elongation of keeping lower temperatures is presented in Table 6.

Table 6 shows the test results means by the sum of time counted from Figure 4 in 100–150 °C function values. Presented data shows differences in reaching set temperatures. The time values rise with increasing contain CSA in structure.

### 3.2. Differential Scanning Calorimetry Results

Differential scanning calorimetry (DSC) was used to confirm the results obtained from the furnace test. The advantages of this form of thermal testing are a smaller mass of samples than in a furnace test, the possibility of comparison with data standards and the value of enthalpy. The disadvantage is a general overview of sample behavior during tests. The DSC tests were conducted in Netzsch DSC 204 station. Tests were conducted in aluminum crucibles. The parameters of the tests are presented in Table 7.

The obtained DSC results confirmed results obtained in the furnace test. The sample Z1 was characterized by two endothermic reactions with peaks at 139.2 °C and 180.8 °C, which were ettringite and C-S-H dehydration; both reactions are close to each other. The next peak was observed at 491.2 °C with a disturbance at the end of the process. Z2 had two peaks in 153.5 °C 1.917 mW/mg and 198.5 °C 1.331 mW/mg. Higher content of ettringite increased the first peak than in the previous sample. More visible is also the second peak, where differences were 0.587 mW/mg for the first and 0.7599 mW/mg for the second. The third peak of Z2 is much smaller than for Z1. The difference is 0.1245 mW/mg. The third sample, Z3, had intensive the first peak 149.1 °C 2.835 mW/mg, which was 34% more intense than Z2, but the second peak moved to 262.7 °C 0.69 mW/mg with almost 50% lower intensity than Z2. In paper [53], the second peak can be connected with the amorphous alumina trihydroxide (AH3) phase occurring, in which the origin is calcium sulfoaluminate cement hydration protocol. In accordance with previous samplers, a peak around 500 °C was absent, which means that portlantide phase was absent in this sample. In Z4, ettringite peak 154.5 °C was higher than in previous samples and was 3.045 mW/mg. It is also at the Z3 trajectory of peaks. The second peak, 269.7 °C, 0.7547, was more intense than Z3 and moved into right. Sample Z5 had the most intense first peak and was 3.264 mW/mg at 171.8 °C. This moved rightmost of all. The second peak was also the most intensive and was 1.005 mW/mg. This was connected with alumina trihydroxide (AH3) phase decomposition [54]. The analysis was made in Proteus analysis software. The DSC curves in Figure 5 were analyzed, and the results are shown in Table 8 to provide more information about phenomena.

The first peak for all materials has changed, respectively, from 107.9 °C for the Z1 sample to 118.7 °C for the Z5 sample. Enthalpy increased with increasing the CSA content in the material, which means that the first peak is connecting with ettringite contains in structure. The sum of the presented enthalpy increased with increasing CSA content in structure from 246.3 J/g to 647.3 J/g.

### 3.3. Flexural and Compressive Strength Parameters

Samples were tested at Servo plus evolution strength machine according to PN-EN 1015-11 in Department of Building Physics and Building Materials, Lodz University of Technology. Samples were tested when they meet the requirements of standard shapes. If samples lose their integrity after 60 min of heat conditioning, then they were not tested like G1 and Z1 60 min, as shown in Figure 6a,b.

#### 3.3.1. Flexural Strength

Test results shown in Figure 7 and detailed in Table 9 confirmed a drop in flexural strength of types of cement and gypsum-based ceramics under high-temperature exposure. Fifteen minutes of exposure caused a 92.1% plunge in flexural strength of the gypsum samples. In 15 and 30 min, the flexural strength of Z1 composites was higher than samples contain CSA. In 60 min, the flexural strength was the highest for Z2, Z5 material, where Z5 and Z3 had similar parameters.

#### 3.3.2. Compressive Strength

In according to presented data in Figure 8 and Table 10, gypsum had the worst compressive strength from measured materials. Under high temperatures, the compressive strength rapidly dropped at 15 min point. After 30 min, the samples had 10.7% of their starting strength. At 60 min, the gypsum was totally damaged.

The highest compressive strength material was only one CSA-based cement. Ceramics containing CSA (Z2 not including) had better compressive strength than OPC. Fifteen minutes in the furnace caused a 40% compressive strength drop in cementous composition. The best result was the Z5 composition at 15 min. The rest of the cement material was around 21 ± 1.5 MPa. Thirty minutes showed better strength of the Z1 and Z2 samples than the Z3–Z5 samples. After 60 min, the best results were for materials containing CSA cement in the matrix. The Z1 samples did not meet the standard requirements of shape. It needs to be mentioned that strength results were in a complex state of strength caused by the presence of hydrated and dehydrated phases. The test results showed a relation between used materials but did not determine calculation strength factors.

The relation between flexural and compressive strength factors are presented in Figure 9. The data axes are reversed to emphasize the declining strength factors during exposure to high-temperature over time. Where FS ratio = flexural strength/compressive strength. Table 11 shows the predicted trend curves based on results from previous strengths tests.

Figure 10 presents the flexural/compressive strength ratios (FS ratio). It needs to be mentioning that the presented results are in time function during exposure on high-temperature. During these test materials, strengths decreased in time, as shown in Figure 9. The highest flexural/compressive strength ratio was for a gypsum-reinforced blend at 0 min. It had rapidly dropped from 0.34 to 0.12 in the first 15 min under fire-temperature load. Next, the ratio increased to 0.17 in 30 min to fall to 0 in 60 min. The Z1 flexural/compressive strength ratio was equal to 0.11. The value of it increased to 0.13 in 15 min and stay at the same level to 30 min. FS ratio dropped down to 0 in 60 min. It shows that in the Z1 samples, the flexural strength during fire decreased slower than compressive strength in 30 min. After this time, flexural and compressive strength rapidly dropped down. Z2 had a similar mechanism of ratio flow. Starting at 0 min points and ending at 30 min, the FS ratio was 0.10. Next, it slowly decreased to a value of 0.08 after 60 min. This shows that flexural strength and compressive strength decreased comparably during high-temperature elevation. FS ratio of Z3 samples had fluctuating character during exposure on high-temperature. In the first 15 min, flexural strength decreased faster than compressive from 0.13 FS ratio to 0.09; in the next 15 min, behavior diverted from 0.09 FS ratio to 0.15. The last 30 min showed that the FS ratio decreases to 0.10. In the Z4 samples in which the FS ratio slightly decreased in the first 15 min, in the next 45 min, the FS ratio rose. This meant that compressive strength decreased more than flexural strength. The last samples, Z5, has increased the FS ratio in all time. FS ratio rises from 0.05 to 0.11 in 60 min, which means that the flexural strength decreased slower than compressive strength during high-temperature.

Table 12 shows the comparison of FS ratio trend curves in time function. Combining results equation from Table 11 and Table 12 together may give information about predict strength factors in chosen time. For most samples, the R^2^ coefficient is higher than 0.96. In the G1, case, it is moderate; and in Z3 cases, it is low-level.

## 4. Conclusions

In this paper, the differences between reinforcement gypsum blend and different cement ratio reinforcement ceramics under high temperatures were showed. Samples were tested according to standards after different heat treatment times. Results contain researched core response on fire-temperature, strength parameters and visual behavior of samples after cooling in the laboratory environment. The measured times correspond to standard EI 15, 30, 60 Fire Resistance class. Conducted research is important from a fire safety point of view. The samples compared from different materials give a general overview of the differences between these materials. It was shown that there are differences between gypsum, OPC and CSA ceramics-based. Research methods based on this same volume of samples, test parameters and the persons who were responsible for test procedures gave elimination of systematic errors. According to the presented papers, the conclusions are as follows:
(1)**Exposure to high-temperature:** Post heat-treatment cooling process and time in ambient air temperature may influence to support cracking effect in sample structures. Thermal shock can appear in water from firefight action or air stream from passive fire systems like SHEV fire systems. Tests prove the cooling properties of gypsum with effectiveness for 30 min in presented volume under fire-temperature load. In gypsum, material strength decreases during the continued cooling effect. After 30 min of heat-treatment, strength is less than 10% of the original. Reinforcement cement blends based on OPC CEM I had a similar cooling effect to gypsum to 200 °C in 35 min, but his strength properties were higher. OPC CEM I matrix modification by CSA cement has a significant influence on time elongation of temperature increase. The effect is connected with the endothermic reaction of ettringite and monosulfite dehydration. In the presented sample volume and their position, every 25% increase containing CSA in the OPC matrix influenced 15 ± 2 min time elongation. These are showing that there is space for improvement in passive fire protection systems. Sixty minutes of exposure on fire-temperature was not beneficial for gypsum and all cement-based material; however, CSA modification in the OPC matrix had a positive influence on ceramics materials. Cement material contains CSA cement had better cooling and strength parameters than gypsum in all cases. The spalling effect was not observed during tests.(2)**Differential scanning calorimetry:** Increasing CSA content increases the total sum of enthalpy of endothermic reactions. Furthermore, the total amount of ettringite is responsible for this behavior. Thermal scanning calorimetry has shown visible forming AH3 phase starting of 50% CSA contribution, and peak intensity increases with CSA contain. DSC results confirm cooling behavior during high-temperature in the furnace. Combining both tests, the conclusion is that as long as the endothermic transformation takes place, the temperature at the measuring point will not rise.(3)**Flexural and compressive strength test:** Reinforcement cement blends contain CSA had higher compressive strength (not including Z2) and lower flexural strength (not including Z3) than reinforcement blend on 100% OPC based in the green state. The flexural strength and compressive strength were higher for composites contain CSA in the matrix during 30 min exposition on high-temperature. Interesting results show FS ratio analysis. Samples contain 100%, and 75% OPC has decreased the FS ratio. It means that flexural strength dropped faster than compressive strength. Samples Z3 with 50% OPC contains had fluctuating FS ratio flow. Samples with 25% and 0% OPC have increased FS ratio during high-temperature. It means that flexural strength decreases slower than compressive strength.

Presented research helped to understand better mechanisms responsible for cooling behavior of cooling mechanism based on cement materials. As shown in the paper, phase selection has a significant influence on cement-based materials during high-temperature exposure. Better understanding phenomena occurring within the materials are needed more tests with different ingredients ratio and load conditioning.

## Figures and Tables

**Figure 1 materials-13-05230-f001:**
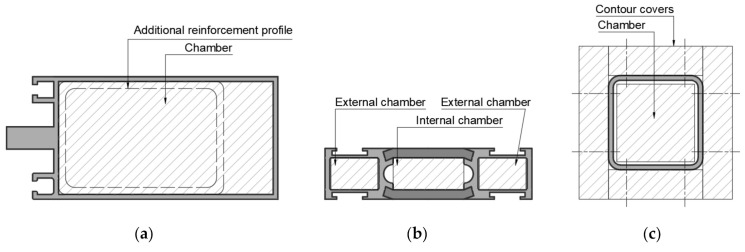
Example of fire protection infills and contour covers: (**a**) Façade profile; (**b**) thermal break window/door/partition profile; (**c**) construction hollow profile.

**Figure 2 materials-13-05230-f002:**
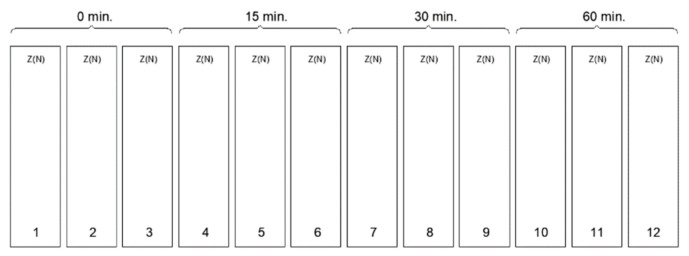
Samples arranged after high-temperature exposure.

**Figure 3 materials-13-05230-f003:**
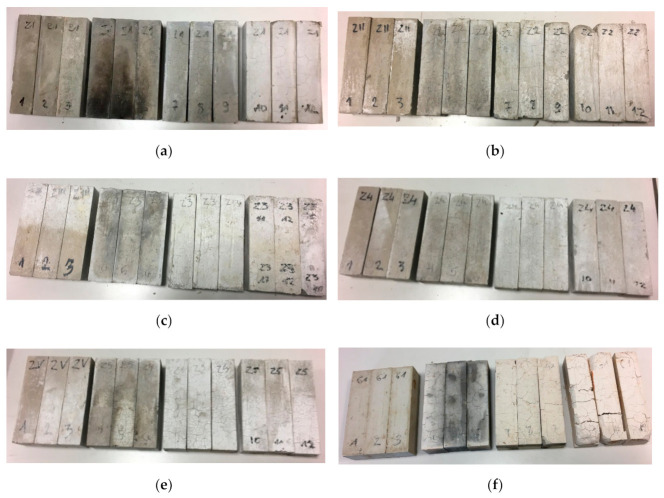
Grouped specimens. (**a**) Z1 specimens; (**b**) Z2 specimens; (**c**) Z3 specimens; (**d**) Z4 specimens; (**e**) Z5 specimens; (**f**) G1 specimens.

**Figure 4 materials-13-05230-f004:**
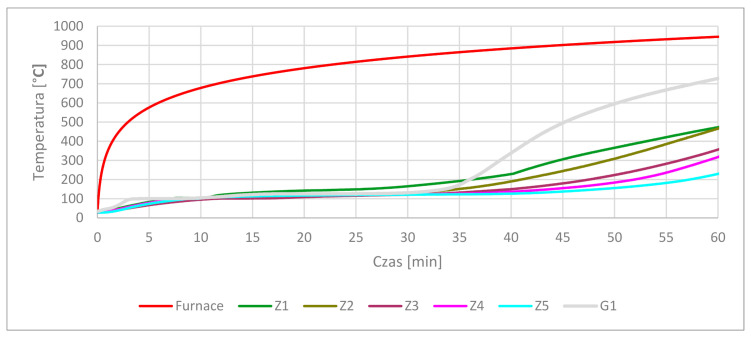
Core temperature results.

**Figure 5 materials-13-05230-f005:**
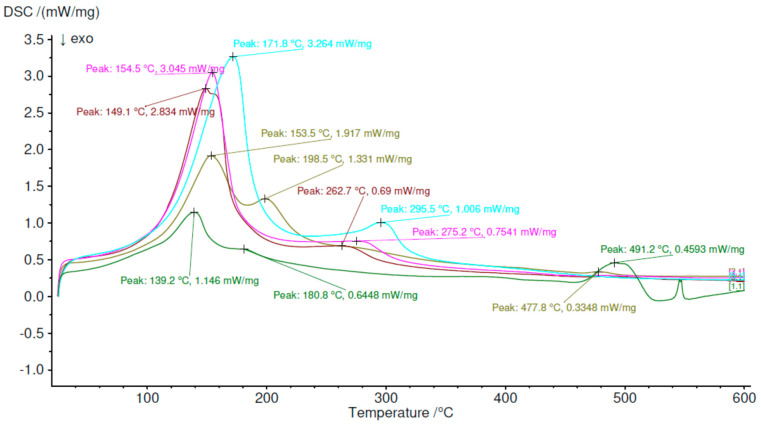
Differential scanning calorimetry results.

**Figure 6 materials-13-05230-f006:**
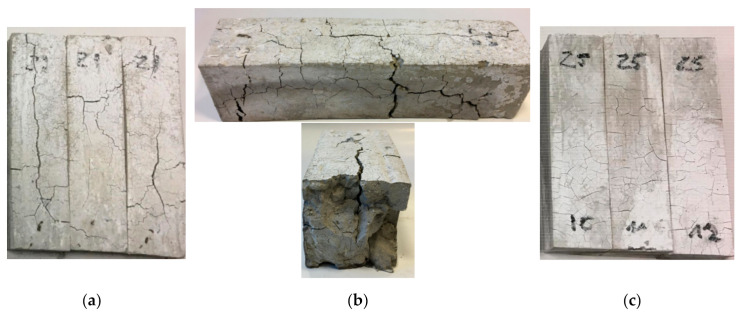
(**a**) Z1 samples after 60 min in the furnace; (**b**) Zoom on Z1 sample after 60 min in the furnace; (**c**) Z5 samples after 60 min in the furnace.

**Figure 7 materials-13-05230-f007:**
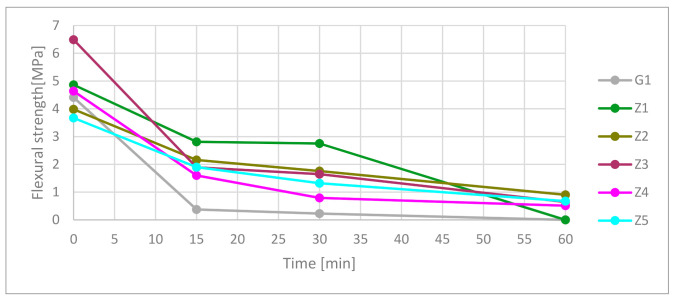
Flexural strength in time function.

**Figure 8 materials-13-05230-f008:**
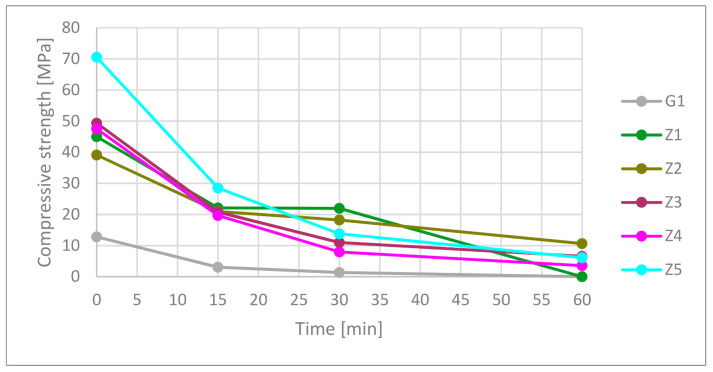
Compressive strength in time function.

**Figure 9 materials-13-05230-f009:**
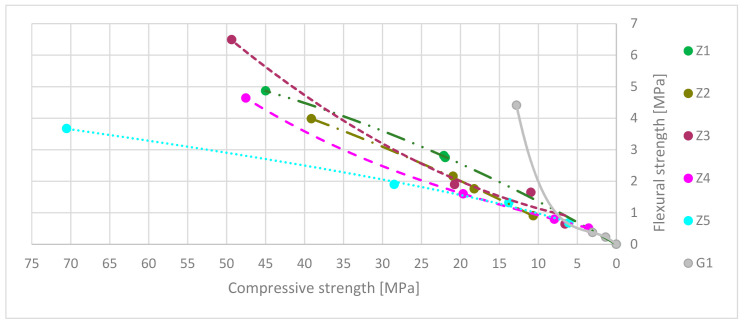
Flexural strength to compressive strength ratio trend curves.

**Figure 10 materials-13-05230-f010:**
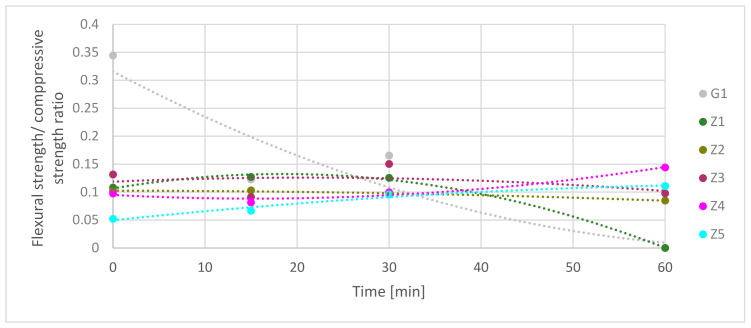
Comparison of flexural strength/compressive strength ratio in time function.

**Table 1 materials-13-05230-t001:** Building Research Institute of Poland guidelines [27].

Temperature Exposure	Phase Decompositions in OPC Concrete in 20–800 °C Range
to 200 °C	Partially ettringite dehydration and beginning calcium–silicate hydrates C-S-H dehydration
to 300 °C	Dehydration end of ettringite and dehydration follow-up of calcium–silicate hydrates C-S-H
to 600 °C	Follow-up dehydration of calcium–silicate hydrates C-S-H and dehydroxylation of portlandite Ca(OH)_2_
to 800 °C	Follow-up dehydration of calcium–silicate hydrates C-S-H and partially carbonate decomposition

**Table 2 materials-13-05230-t002:** Calcium sulfoaluminate (CSA) phase decomposition.

Temperature Exposure	Phase Decompositions in CSA Concrete
from 90 °C	Ettringite dehydration and decomposition to monosulfite and calcium sulfate
from 150 °C	Partially monosulfite dehydration
200–230 °C	Alumina trihydrate dehydroxylation
from 450 °C	Monosulfite dehydration

**Table 3 materials-13-05230-t003:** Summary of cement phase decomposition.

Temp. [°C]	Phase	Mark
50–100	${\mathrm{CaSO}}_{4}\cdot 2H_{2}O$	${C\bar{S}H}_{2}$
100–150	${\mathrm{CaSO}}_{4}\cdot 2H_{2}O$	${C\bar{S}H}_{2}$
	${\mathrm{CaSO}}_{4}\cdot \frac{1}{2}H_{2}O$	$C\bar{S}H_{0.5}$
	-	C−S−H ^a^
	${\mathrm{Ca}}_{3}{\mathrm{Al}}_{2}O_{6}\cdot 3{\mathrm{CaSO}}_{4}\cdot 32H_{2}O$ ^a,b^	$C_{3}A\cdot 3C\bar{S}\cdot H_{32}$ ^a,b^
150–200	${\mathrm{CaSO}}_{4}\cdot 2H_{2}O$	${C\bar{S}H}_{2}$
	${\mathrm{CaSO}}_{4}\cdot \frac{1}{2}H_{2}O$	${C\bar{S}H}_{0.5}$
	${\mathrm{Ca}}_{3}{\mathrm{Al}}_{2}O_{6}\cdot {\mathrm{CaSO}}_{4}\cdot 12H_{2}O$	$C_{3}A\cdot C\bar{S}\cdot H_{12}$
200–250	${\mathrm{CaSO}}_{4}\cdot \frac{1}{2}H_{2}O$	${C\bar{S}H}_{0.5}$
300–350	${\mathrm{Ca}}_{3}{\mathrm{Al}}_{2}O_{6}\cdot 6H_{2}O$	$C_{3}{\mathrm{AH}}_{6}$
450–500	${\mathrm{Ca}}_{3}{\mathrm{Al}}_{2}O_{6}\cdot {\mathrm{CaSO}}_{4}\cdot 12H_{2}O$ ^d^	$C_{3}A\cdot C\bar{S}\cdot H_{12}$ ^d^
	$\mathrm{Ca}{\left(\mathrm{OH}\right)}_{2}$ ^a^	CH ^a^
	${\mathrm{Ca}}_{3}{\mathrm{Al}}_{2}O_{6}\cdot 6H_{2}O$ ^d^	$C_{3}{\mathrm{AH}}_{6}$ ^d^

Important: a—highest lost, b—main lost, d—low lost.

**Table 4 materials-13-05230-t004:** Ingredients compounds (g).

N^o^	Name	Gypsum	OPC	CSA	Water	Glass Fiber
1	G1	4400	-	-	2640	88
2	Z1	-	4400	-	2200	88
3	Z2	-	3300	1100	2200	88
4	Z3	-	2200	2200	2200	88
5	Z4	-	1100	3300	2200	88
6	Z5	-	-	4400	2200	88

**Table 5 materials-13-05230-t005:** Density (kg/m^3^).

		Time			
	Marks	0	15	30	60
Density kg/m^3^	Z1	1872.4	1782.6	1351.6	1242.2
	Z2	1849.0	1468.8	1339.8	1248.7
	Z3	1790.4	1464.9	1341.1	1216.1
	Z4	1761.8	1450.5	1266.9	1157.5
	Z5	1812.5	1506.5	1355.5	1218.7
	G1	1182.3	1018.2	990.9	921.9

**Table 6 materials-13-05230-t006:** Elongation time in comparison of cement core temperature increase in 100–150 °C interval.

Samples	Z1	Z2	Z3	Z4	Z5
Time [min]	19:08	26:30	29:49	36:28	40:28
Rise	Ref	+39%	+56%	+91%	+111%

**Table 7 materials-13-05230-t007:** Parameters of differential scanning calorimetry (DSC) tests.

Specimen	Color	Mass [mg]	Range	Atmosphere
Z1		16.5	25 °C/15.0 (K/min)/600 °C	N_2_, 20.0 mL/min
Z2		17.3	25 °C/15.0 (K/min)/600 °C	N_2_, 20.0 mL/min
Z3		16.0	25 °C/15.0 (K/min)/600 °C	N_2_, 20.0 mL/min
Z4		14.3	25 °C/15.0 (K/min)/600 °C	N_2_, 20.0 mL/min
Z5		16.8	25 °C/15.0 (K/min)/600 °C	N_2_, 20.0 mL/min

**Table 8 materials-13-05230-t008:** DSC analysis.

N^o^	Peak	T_s_ [°C]	T_d_ [°C]	Heat Flow [mW/mg]	Enthalpy ∆H [J/g]	$\overset{n}{\underset{i=3}{\sum }}∆H_{i}$ [J/g]
Z1	Peak 1	107.9	139.2	1.146	141.49	246.30
	Peak 2	165.0	180.8	0.644	37.61	
	Peak 3	466.7	491.2	0.4593	67.20	
Z2	Peak 1	112.0	153.5	1.917	239.08	336.76
	Peak 2	183.0	198.5	1.331	95.52	
	Peak 3	452.3	477.8	0.3348	2.16	
Z3	Peak 1	113.4	149.1	2.834	446.04	495.60
	Peak 2	240.1	262.7	0.69	49.56	
Z4	Peak 1	114.7	154.5	3.045	489.48	568.80
	Peak 2	234.3	275.2	0.7541	79.32	
Z5	Peak 1	118.7	171.8	3.264	547.92	647.30
	Peak 2	264.4	295.5	1.006	99.38	

T_s_—initial phase transition temperature, T_d_—maximal process temperature.

**Table 9 materials-13-05230-t009:** Flexural strength in time function.

Time [min]	Flexural Strength [MPa]					
	G1	Z1	Z2	Z3	Z4	Z5
0	4.41	4.87	3.98	6.49	4.64	3.67
15	0.37	2.81	2.16	1.89	1.60	1.90
30	0.23	2.75	1.76	1.65	0.79	1.32
60	0.00	0.00	0.90	0.64	0.51	0.68

**Table 10 materials-13-05230-t010:** Compressive strength in time function.

Time [min]	Compressive Strength [MPa]					
	G1	Z1	Z2	Z3	Z4	Z5
0	12.81	45.00	39.12	49.36	47.53	70.53
15	3.05	22.13	20.94	20.76	19.65	28.50
30	1.37	21.95	18.22	10.97	7.95	13.79
60	0.00	0.00	10.65	6.57	3.55	6.08

**Table 11 materials-13-05230-t011:** Comparison of flexural strength and compressive strength ratio trend curves and coefficient of determination.

Name	Trend Curve Equation	R^2^ Value
G1	f_f_ = 0.0042f_c_^3^ − 0.0441f_c_^2^ + 0.2179– 9 × 10^−13^	1
Z1	f_f_ = −0.0008f_c_^2^ + 0.1437f_c_– 8 × 10^−5^	0.9999
Z2	f_f_ = −0.0006f_c_^2^ + 0.1368f_c_ − 0.4973	0.9992
Z3	f_f_ = 0.0017f_c_^2^ + 0.02359f_c_ + 0.6085	0.9849
Z4	f_f_ = 2 × 10^5^f_c_^3^ − 0.0002f_c_^2^ + 0.0645f_c_ + 0.2842	1
Z5	f_f_ = 0.207f_c_^0.6752^	0.9932

**Table 12 materials-13-05230-t012:** Comparison of flexural strength (FS) ratio in time function equations and coefficient of determination.

Name	f(t) = FS Ratio	R^2^ Value
G1	f(t) = 6 × 10^−5^t^2^ − 0.0086t + 0.3159	0.8374
Z1	f(t) = −8 × 10^−5^t^2^ + 0.0028t + 0.1065	0.9971
Z2	f(t) = −5 × 10^−6^t^2^– 9 × 10^−6^t + 0.1025	0.9731
Z3	f(t) = −2 × 10^−5^t^2^ + 0.0007t + 0.1187	0.1507
Z4	f(t) = 3 × 10^−5^t^2^ − 0.0009t + 0.095	0.9612
Z5	f(t) = −1 × 10^−5^t^2^ + 0.0017t + 0.0497	0.9677

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
