# Peer review of "Fire-Temperature Influence on Portland and Calcium Sulfoaluminate Blend Composites"

_materials, 2020, doi:10.3390/ma13225230_

Round 1

Reviewer 1 Report

The following comments and suggestions will improve the understanding of the material presented by this paper.

Review comments

  • The paper needs professional editing. There are several English errors through out the text. For example, in line 23 there are two full-stops, in line 47 it should be written ‘…there are….’, in line 62 it should be written ‘…mechanisms…’, etc.
  • In line 165, in the title the word ‘compression’ should be substituted with the word ‘compressive’.
  • In line 274 in the title the word ‘Summarize’ should be omitted.
  • The authors should explain better the paragraph after Figure 5.
  • In Figure 5 there is a ‘knee’ at the curves at certain minutes (for example in Z1 curve there is a knee at 40 min). Can this be explained?
  • The paragraph 212-216 has to be explained better.
  • An additional Figure, like Figure 7, should be added with Z5 samples.
  • It would be beneficial to add (if the authors have) some data without the compounds Gypsum and CSA.
  • Attention should be given at conclusions because there are several English mistakes.

Author Response

Dear Reviewer

Please see the attachment. Best regards Konrad A. Sodol.

Reviewer 2 Report

Fire-temperature influence on Portland and calcium sulfoaluminate blend composites

The Paper deal about the fire temperature influence on portland and calcium sulfo-aluminate of several blend composites. Although the papers present an interesting experimental campaign there are some points that needs of clarification. It is suggested to review the keywords recalling to the experimental campaign and to the rheological effect analysed. In addition, the Authors should clarify what is the level of novelty and the innovative contribution of the paper in the field of fire engineering. A significant improvement of the article it is expected in order to consider it for publication.

Lines 23 => (Typo: double dot)

Lines 31 => Please check the grammar form

Line 69 => (Typo: silicate-cement).

Lines 171-172 => Pleas check the form.

Lines 250-260 => It is suggested to interpolate the point of the graphs with a polynomial law to avoid unexpected path due to the reduced number of the specimen. The decreasing ratio of the compression strength and of the flexural strength must be commented in the text.

Lines 274 - 316 => Conclusion it is now a list of generic phrases. Please include an organic and quantitative discussion of the results.  

Lines 285 => Please check the form

Line 330 => Please check the form of the reference. The reference should be in english and must respect the MDPI format.

Author Response

Dear Reviewer,

Reviewer 3 Report

The aim of this paper is to study the fire-temperature influence on Portland and calcium sulfoaluminate blend composites. The introduction could be improved with reading the articles: “An empathetic added sustainability index (EASI) for cementitious based construction materials” Journal of Cleaner Production, 220 (2019), 475-482.

The comments have been included in the pdf.

Author Response

Dear Reviewer,

Best Regards

Reviewer 4 Report

This study is an interesting paper reviewing the results of the fire resistance performance according to the blending ratio of gypsum, OPC and CSA. Because there is no review on the unit price and only the performance and results of the test specimens have been reviewed, it seems that there will be difficulties in practical use.

However, in terms of securing basic data on the fire resistance performance of CSA, it is considered to be of utility value.

In order to be considered for this paper to be published, it seems that a major revision is required as follows.

  1. Please correct the error part of the overall paper.

        For example:

        - Line 23 : ~ range.. (2 Periods)

        - Line 35 : object, building (2 Spaces) etc.

  1. Why are W/G(=0.6) and W/C(0.5) set to different ratios in Table 4? And by what standards did you add glass fiber? Please explain these.

  1. Is "(Line 178) Density increase after high-temperature is an expected phenomenon." correct? Please check if the meaning has been reversed.

  1. (Fig. 8~9) Have you analyzed the properties of the ratio of flexural strength/compression strength for each specimen? If this is analyzed, there may be interesting results.

  1. (Line 182~194) More interesting results will be derived if the amount of crack generation can be quantitatively evaluated for each specimen. Have you ever thought about this?

  1. If “3. Results and discussion” is restructured, the structure of this study will become clear. For example,

        3.1 Core heat curves

        3.2 Differential scanning calorimetry

        3.3 Flexural and compression strength

  1. The "4. Summarize and conclusions" section is too long and unstructured. Overall, please rewrite and structure more concisely referring to comment NO.6.

  1. In the conclusion, please reinforce the limitations of this study and future research directions.

Author Response

Dear Reviewer

Best Regards

Round 2

Reviewer 3 Report

The authors modified the paper according to the reviewer's instructions. 

Reviewer 4 Report

Overall, the content has been improved by reflecting many of the comments I gave. However, it is necessary to submit a specific answer for each reviewer comment. The authors presented only the text that indicated the revised content of the manuscript.